# Training Spiking Neural Networks with Event-driven Backpropagation

**Yaoyu Zhu**[1], **Zhaofei Yu**[1,2*], **Wei Fang**[1], **Xiaodong Xie**[1], **Tiejun Huang**[1], **Timothée Masquelier**[3]
[1]School of Computer Science, Peking University
[2]Institute for Artificial Intelligence, Peking University
[3]Centre de Recherche Cerveau et Cognition (CERCO), UMR5549 CNRS - Univ. Toulouse 3,
Toulouse, France

## Abstract

Spiking Neural networks (SNNs) represent and transmit information by spatiotemporal spike patterns, which bring two major advantages: biological plausibility and suitability for ultralow-power neuromorphic implementation. Despite this, the binary firing characteristic makes training SNNs more challenging. To learn the parameters of deep SNNs in an event-driven fashion as in inference of SNNs, backpropagation with respect to spike timing is proposed. Although this event-driven learning has the advantages of lower computational cost and memory occupation, the accuracy is far below the recurrent neural network-like learning approaches. In this paper, we first analyze the commonly used temporal backpropagation training approach and prove that the sum of gradients remains unchanged between fully-connected and convolutional layers. Secondly, we show that the max pooling layer meets the above invariance rule, while the average pooling layer does not, which will suffer the gradient vanishing problem but can be revised to meet the requirement. Thirdly, we point out the reverse gradient problem for time-based gradients and propose a backward kernel that can solve this problem and keep the property of the invariable sum of gradients. The experimental results show that the proposed approach achieves state-of-the-art performance on CIFAR10 among time-based training methods. Also, this is the first time that the time-based backpropagation approach successfully trains SNN on the CIFAR100 dataset. Our code is available at https://github.com/zhuyaoyu/SNN-event-driven-learning.

## 1 Introduction

Motivated by the principles of brain computing, Spiking Neural Networks (SNNs) are considered as the third generation of neural networks [1, 2]. SNNs are developed to work in power-critical scenarios, such as edge computing. When run on dedicated neuromorphic chips, they can accomplish the tasks [3, 4, 5] with ultra-low power consumption [6, 7, 8, 9, 10, 11, 12]. In contrast, the last generation of neural networks – Artificial Neural Networks (ANNs) [13], generally require a large amount of computation resource (e.g., GPUs). This advantage of SNNs on power consumption largely relies on efficient event-based computations [14, 15]. Another advantage of SNNs originates from their biological reality (compared to ANNs). The similarity between SNNs and biological brains provides an excellent opportunity to study how the brain computes at the neuronal circuit level [16].

Compared with artificial neural networks, developing supervised learning algorithms for spiking neural networks requires more effort. The main challenge for training SNNs comes from the binary nature of spikes and the non-differentiability of the membrane potential at spike time. This difficulty

---

*Corresponding author

36th Conference on Neural Information Processing Systems (NeurIPS 2022).

in training impedes the performance of SNNs in pattern classification tasks compared to their ANN counterparts. Existing supervised learning methods of SNNs can be grouped into two categories:

The first category consists of recurrent neural network (RNN)-like learning algorithms. These algorithms treat spiking neural networks as binary-output recurrent neural networks and handle the discontinuities of membrane potential at spike times with continuous surrogate derivatives [17]. They typically train deep SNNs with surrogate gradients based on the idea of backpropagation through time (BPTT) algorithm [18, 19, 20, 21, 22, 23, 24, 25, 26, 27, 28]. While competitive accuracies are reported on the MNIST, CIFAR-10, and even ImageNet datasets [29, 30, 31], the gradient information is propagated each time step, whether or not a spike is emitted (as shown in Fig. 1). Therefore, these approaches do not follow the event-driven nature of spiking neural networks, which lose the asynchronous characteristic of SNNs and consume much power when trained on neuromorphic hardware.

The second category is event-driven algorithms, which propagate gradient information through spikes. Precise spiking timing acts an important role in this situation, and they are extensively used in such algorithms [32, 33, 34, 35, 36, 37, 38, 39]. Classical examples include SpikeProp [32] and its variants [33, 40, 41]. These algorithms approximate the derivative of spike timing to membrane potential as the negative inverse of the time derivative of membrane potential function. This approximation is actually mathematically correct without preconditions [42]. Some other works apply non-leaky integrate-and-fire neurons to stabilize the training process [35, 38, 43]. Most of these works restrict each neuron to fire at most once, which inspires [44] to take the spike time as the state of a neuron, and model the relation of neurons by this spike time. As a result, the SNN is trained similarly to an ANN. Among the methods trained in an event-driven fashion (not modelling the relation of spike time to train like ANNs), the state-of-the-art model is TSSL-BP [39]. However, they use RNN-like surrogate gradients (a sigmoid function) to assist training. Hence, it is still challenging to train SNNs in a pure event-driven fashion.

In this work, we develop a novel event-driven learning algorithm that can train high-performance deep SNNs. The main contributions of our work are as follows:

1. We prove that the typical SNN temporal backpropagation training approach assigns the gradient of an output spike of a neuron to the input spikes generating it. After summing this assignment rule altogether, we find that the sum of gradients is unchanged between layers.

2. We analyze the case of the pooling layer (which does not have neurons) and find that average pooling does not keep the gradient sum unchanged, but we can modify its backward formulas to meet the requirement. Meanwhile, the max-pooling layer satisfies the rule initially.

3. We point out the reverse gradient problem in event-driven learning that the direction of the temporal gradient is reversed during backpropagation when the kernel function of an input spike is decreasing. Then we propose a backward kernel function that addresses this problem while keeping the sum of gradients unchanged between layers.

4. The adjusted average pooling layer and the non-decreasing backward kernel enhances the performance of our model as well as the convergence speed. To our best knowledge, our proposed approach achieves state-of-the-art performance on CIFAR10 among event-driven training methods (with temporal gradients) for SNNs. Meanwhile, our method is the first event-driven backpropagation approach that successfully trains SNN on the larger-scale CIFAR100 dataset.

## 2 Backgrounds and Related Work

The gradient-based learning of spiking neural networks contains two stages: the forward (inference) and the backward (learning) stages. In the forward stage, Leaky Integrate-and-Fire (LIF) neurons are most commonly used [18, 21, 26, 39], while other types of neurons are also applicable [32, 35]. Typically, these neuron models can be changed to the form of the Spike Response Model (SRM) [37, 45, 46], which is easily represented in an event-driven fashion.

In the backward stage, the methods used by existing works exhibits more diversity. Here, we classify existing approaches from two dimensions: whether non-spike information is needed in discrete time steps (RNN-like) or not (event-driven) and whether the gradient represents spike scale (activation-based) or spike timing (time-based).

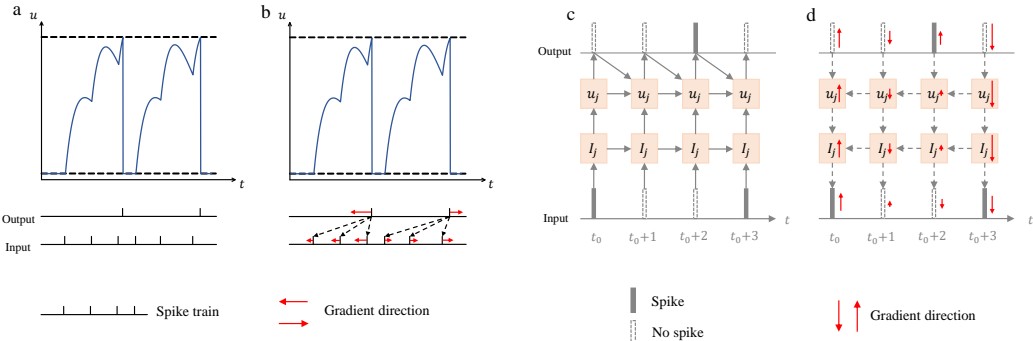

Figure 1: (a) The input and output spike train, as well as membrane potential curve of a neuron in event-driven learning with (b) time-based gradient: When the spiking neural network is simulated in an event-driven fashion, a neuron updates its state when an input spike arrives or it emits an output spike. The gradients here indicate whether spikes should move leftward or rightward along the time axis. (c) The input, output, membrane potential sequence of a neuron in RNN-like learning with (d) activation-based gradient: The spiking neural network is forced to be simulated in discrete time steps (since gradient should be propagated in non-spike time steps) and the gradients denote whether spikes should be 'larger' or 'smaller'.

**Event-driven learning v.s. RNN-like learning:** In both forward and backward computation of event-driven learning, information is only carried by spikes in SNNs. Specifically, in backward computation, gradient information is propagated through spikes [32, 33, 41, 35] (shown in Fig. 1a-b). On the other side, in RNN-like learning, information is not only carried by spikes in backward computation. Especially, gradient information can be propagated through a neuron that does not emit a spike in backward computation (shown in Fig. 1c-d). This gradient propagation is achieved by a surrogate function [12, 17, 18, 23, 47], which is a function of the membrane potential at the current time step $u_t$, and the firing threshold $\theta$.

**Time-based gradient v.s. activation-based gradient:** Time-based gradients represent the (reverse) direction that the timing of a spike should move, that is, to move leftward or rightward on the time axis [32]. In backward propagation, the derivative of the firing time of a spike to the corresponding membrane potential $\frac{\partial t}{\partial u}$ is often approximated as $\frac{-1}{\frac{\partial u}{\partial t}}$ [32, 33], denoting how the change of membrane potential will change the spike firing time (Fig. 1b). On the other side, activation-based approaches replace the Heaviside neuron activation function $\Theta(\cdot)$ (spike $s_t = \Theta(u_t - \theta)$) in forward propagation with derivable functions $\sigma(\cdot)$ in backward propagation, whether there are spikes in the current time step [18, 26, 31, 21]. Therefore, activation-based approaches essentially regard SNNs as binary RNNs and train them with approximated gradients, where the gradients indicate whether the values in the network (including the binary spikes) should be larger or smaller (Fig. 1d).

As a result, time-based gradients are event-driven by nature, since the temporal gradient could only be carried by spikes. Meanwhile, activation-based gradients are more suitable for the RNN-like training scheme since the diversity of surrogate gradients largely relies on the fact that $u_t \neq \theta$ in discrete time steps [17], which no longer holds in continuous time simulation. If we want to apply activation-based gradients to event-driven learning, there should only be one value $\frac{\partial s}{\partial u}$ when the membrane potential reaches the threshold.

Tab. 1 lists whether a gradient type can be used in a learning fashion. It should be noticed that although activation-based gradient is more suitable for RNN-like learning, it is still able to be used for event-driven learning.

Table 1: Whether the gradient type can be used in a learning fashion

|  | Event-driven learning | RNN-like learning |
|---|---|---|
| Time-based gradient | ✓ | ✗ |
| Activation-based gradient | ✓ | ✓ |

# 3 Methods

## 3.1 Forward Formulas

We use the spike response model [1] for neurons in the network. The forward propagation in the network can be described as follows:

$$u_i^{(l)}(t) = \int_{t_{i,last}^{(l)}}^{t} \left( \sum_j w_{ij}^{(l)} \cdot s_j^{(l-1)}(\tau) \right) \cdot \epsilon(t - \tau) d\tau, \tag{1}$$

$$s_i^{(l)}(t) = \delta(u_i^{(l)}(t) - \theta). \tag{2}$$

Here $u_i^{(l)}(t)$ denotes the membrane potential of neuron $i$ in layer $l$ at time $t$, $w_{ij}^{(l)}$ denotes the weight between neuron $j$ in layer $l-1$ and neuron $i$ in layer $l$. $t_{i,last}^{(l)}$ is the time of last spike of neuron $i$ in layer $l$, and $s_i^{(l)}(t)$ represents the spike emitted from neuron $i$ at time $t$. The function $\delta(\cdot)$ is the Dirac Delta function and $\theta$ is the firing threshold. The spike response kernel $\epsilon(t)$ can be described by

$$\epsilon(t) = \frac{\tau_m}{\tau_m - \tau_s}(e^{-\frac{t}{\tau_m}} - e^{-\frac{t}{\tau_s}}), \tag{3}$$

where $\tau_m$ and $\tau_s$ are the membrane time constant and the synapse time constant respectively. Notice that we do not use reset kernels as in previous works [39, 41]. Instead, we eliminate the influence of input spikes prior to the last output spike on membrane potentials.

## 3.2 Rethinking the Classical Time-based Backward Propagation Formula

In this subsection, we analyze the classical time-based backpropagation formula in SNNs. We first theoretically prove that the backpropagation rule essentially assigns gradients of output spikes of neurons to their input spikes. Then we check the pooling layer and show that the average pooling should be adjusted in backpropagation to satisfy the gradient assignment mechanism, while the max pooling naturally satisfies this mechanism.

**Invariant sum of gradients among layers with weights.** The most commonly used time-based gradient backpropagation method origins from [32]. The two key approximations are as follows:

$$\frac{\partial t_k(s_i^{(l)})}{\partial u_i^{(l)}(t_k)} = \frac{-1}{du_i^{(l)}(t_k)/dt} = \left( \sum_{t_{k,last}(s_i^{(l)}) < t_m(s_j^{(l-1)}) \leq t_k(s_i^{(l)})} w_{ij}^{(l)} \cdot \frac{\partial \epsilon(t_k - t_m)}{\partial t_m} \right)^{-1}, \tag{4}$$

$$\frac{\partial u_i^{(l)}(t_k)}{\partial t_m(s_j^{(l-1)})} = w_{ij}^{(l)} \cdot \frac{\partial \epsilon(t_k - t_m)}{\partial t_m}, \tag{5}$$

where $t_k(s_i^{(l)})$ denotes the firing time $t_k$ of neuron $i$ in layer $l$, $t_{k,last}(s_i^{(l)})$ is the firing time of the last spike emitted by neuron $i$ before time $t_k$. $\frac{\partial t_k(s_i^{(l)})}{\partial \cdot}$ means the influence of changing other variables on the timing of a spike, and $\frac{\partial \cdot}{\partial t_k(s_i^{(l)})}$ is the influence of changing spike timing on that variable. Combining Eqs. 4-5 and the forward formulas, we can get an invariant equality:

$$\sum_j \sum_{t_{k,last}(s_i^{(l)}) < t_m(s_j^{(l-1)}) \leq t_k(s_i^{(l)})} \frac{\partial t_k(s_i^{(l)})}{\partial t_m(s_j^{(l-1)})} = 1. \tag{6}$$

The proof is provided in Appendix. Eq. 6 implies the fact that the reference time ($t = 0$) is meaningless, and only relative spike times matter. If we increase all the spike times in layer $l-1$ by 1 unit along the time axis, then all the spike times in layer $l$ are also increased by 1 unit along the time axis. Denote the loss function as $L$, then the gradient of $L$ with respect to $t_m(s_j^{(l-1)})$ is:

$$\frac{\partial L}{\partial t_m(s_j^{(l-1)})} = \sum_i \sum_{t_m(s_j^{(l-1)}) < t_k(s_i^{(l)}) \leq t_{m,next}(s_j^{(l-1)})} \frac{\partial L}{\partial t_k(s_i^{(l)})} \cdot \frac{\partial t_k(s_i^{(l)})}{\partial t_m(s_j^{(l-1)})}, \tag{7}$$

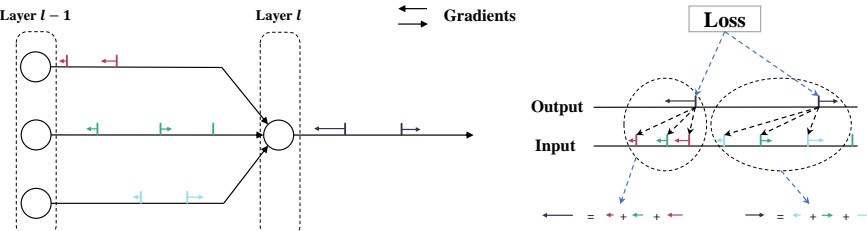

Figure 2: Left: Spikes (vertical ones) and their corresponding gradients (horizontal arrows) in layer $l-1$ and layer $l$. Right: Input and output of the neuron in layer $l$. The input integrates spikes from connected neurons in layer $l-1$ (notice the input is not multiplied by weights). In backpropagation, the gradients are transmitted from the output of the neuron to the inputs. The key property is that the gradient of an output spike **exactly equals** the sum of gradients of input spikes between the last output spike and it.

where $t_{m,next}(s_j^{(l-1)})$ denotes the firing time of the next spike emitted by neuron $j$ after time $t_m$. Therefore, we actually decompose the gradient $\partial L/\partial t_k(s_i^{(l)})$ from layer $l$ into (part of) a set of gradients $\partial L/\partial t_m(s_j^{(l-1)})$ of the last layer $l-1$, and keep their sum unchanged. In other words, we assign the weighted sum $\partial L/\partial t_k(s_i^{(l)})$ by weights $\partial t_k(s_i^{(l)})/\partial t_m(s_j^{(l-1)})$ to the gradients $\partial L/\partial t_m(s_j^{(l-1)})$ in the last layer, as shown in Fig. 2.

If we sum all the gradients together, we can get another invariant in this backpropagation rule:

$$\sum_i \sum_{t_k} \frac{\partial L}{\partial t_k(s_i^{(l)})} = \sum_j \sum_{t_m} \frac{\partial L}{\partial t_m(s_j^{(l-1)})}, \tag{8}$$

which means the sum of gradients $\sum_i \sum_{t_k} \frac{\partial L}{\partial t_k(s_i^{(l)})}$ never changes between layers under this rule.

**Gradient sum invariance for pooling layers.** The above equations determine the gradient propagation in fully-connected and convolution layers (which contain neurons). The case for pooling layers (which do not contain neurons) is illustrated in Fig. 3.

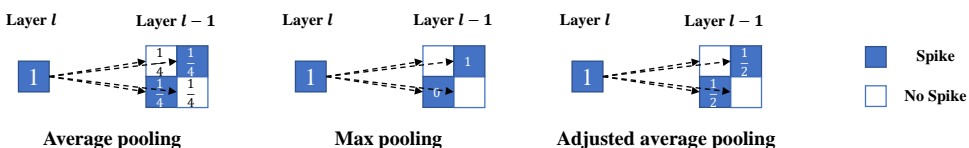

Figure 3: Different pooling strategies. In average pooling (left), the gradients are propagated to neurons not emitting spikes, which will cause gradient vanishing problem. Meanwhile, max pooling (middle) passes all gradients to one of the neurons emitting spikes in the backward stage, keeping the total gradient unchanged. The average pooling can be adjusted (right) to pass gradients to all neurons emitting spikes in the last layer to avoid the gradient vanishing problem.

In average pooling with kernel size $k$, the gradient of one spike (at time $t$) in layer $l$ is averagely propagated to $k \times k$ neurons in layer $l-1$ connected to it. Some of these $k \times k$ neurons may not emit spikes at time $t$. However, the gradients are also propagated to those neurons, which cannot further propagate the gradients to the previous layers. For instance, the two white squares (neurons) in layer $l-1$ in Fig. 3 receive gradients, but they will not further propagate the gradients to layer $l-2$. Thus, a part of the gradients is lost in the backpropagation of the average pooling layer, which might cause the gradient vanishing problem. Meanwhile, the sum of gradients carried by the spikes is not kept among layers in this case. We can adjust the backpropagation stage in average pooling to satisfy the gradient sum invariance requirement by increasing the multiplier from $1/k^2$ to $1/n_{spike}$, where $n_{spike}$ is the number of spikes emitted by the $k \times k$ neurons in layer $l-1$ at the current time step.

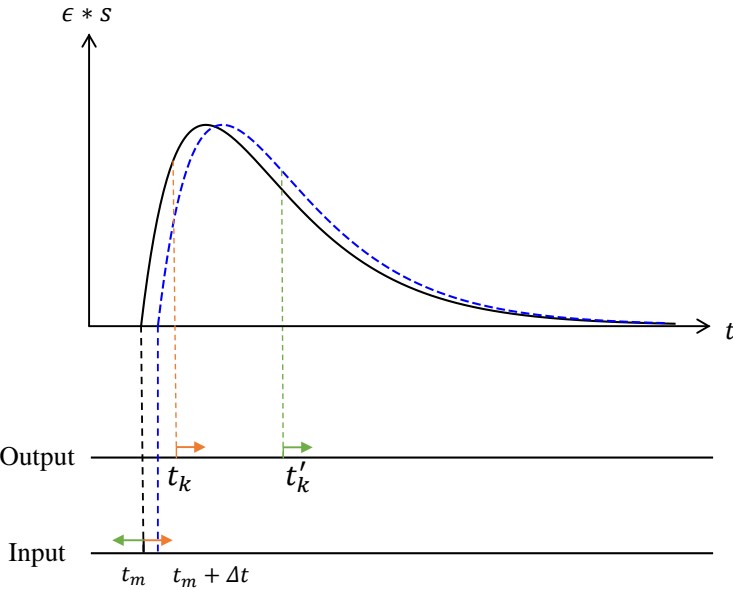

Figure 4: A comparison of temporal gradient direction in backpropagation between two different cases. Consider moving one of its input spike moving from time $t_m$ to $t_m + \Delta t$ for an SRM neuron. Without loss of generality, we assume the weight multiplied on this input spike is positive. If the next output spike is fired at time $t_k$, moving the input spike from $t_m$ to $t_m + \Delta t$ will decrease the membrane potential of this neuron at time $t_k$, which causes this spike to fire later. As a result, the gradient direction from the output spike at time $t_k$ will be kept. In opposite, if the next output spike is fired at time $t'_k$, moving the input spike from $t_m$ to $t_m + \Delta t$ will cause this spike to fire earlier, so the gradient direction from the output spike at time $t'_k$ will be reversed.

On the contrary, in max pooling, the gradient of a spike in layer $l$ is entirely propagated to one of the spikes emitted by its connected neurons in layer $l-1$ (shown in the middle of Fig. 3). This maintains the property of the invariable sum of gradients.

It should be noticed that, although the backpropagation stage of max pooling is different from adjusted average pooling in discrete simulation, they are almost surely the same in the continuous simulation since (almost surely) no two spikes emit at exactly the same time in this case.

### 3.3 Deficiencies of The Typical Time-based Gradient Propagation and A New Approach

In this section, we first point out the reverse gradient problem in event-driven learning: the gradient direction for spike timing gets wrong when the spike response kernel is decreasing. Then we propose a backward kernel that can not only solve the reverse gradient problem but also keep the property of the invariable sum of gradients.

**The reverse gradient problem.** Fig. 4 illustrates the membrane potential response of a neuron (with index $i$) to one of its input spikes (with a positive weight) from presynaptic neuron $j$. As in equation (3), the spike response kernel is a double-exponentail function. Notice that this is not the whole membrane potential of the neuron $i$ as it also receives the inputs from other presynaptic neurons. We consider two spike times ($t_k$ and $t'_k$) of neuron $i$ and one spike time $t_m$ of presynaptic neuron $j$. If the next spike of neuron $i$ fires at time $t_k$, moving the presynaptic spike from $t_m$ to $t_m + \Delta t$ will decrease membrane potential at time $t_k$, which means postponing the spike at time $t_k$ ($t_m \uparrow \Rightarrow u[t_k] \downarrow \Rightarrow t_k \uparrow$). This result shows that if the presynaptic neuron fires earlier, the postsynaptic neuron will fire earlier in this case.

Oppositely, if the next spike of neuron $i$ fires at time $t'_k$, moving the input spike from $t_m$ to $t_m + \Delta t$ will cause a increase of $u[t'_k]$, which further moves $t'_k$ leftward ($t_m \uparrow \Rightarrow u[t'_k] \uparrow \Rightarrow t'_k \downarrow$). As a result, if we want to move the output spike at $t'_k$ leftward, we should move the input spike at $t_m$ rightward, **which reverses the direction**. This might cause a problem: When we want to move $t'_k$ leftward, we **want the neuron to emit more spikes**. However, in gradient backpropagation, it moves $t_m$

rightward (assume weight $w_{ij} > 0$), which may cause the neuron in the last layer to spike fewer, further **causing neurons in the current layer to spike fewer**.

More formally, we assume the neuron $i$ receives a input spike at time $t_m$ from presynaptic neuron $j$ with synaptic weight $w_{ij}$, then the membrane potential of neuron $i$ at time $t_k$ is:

$$u_i(t_k) = w_{ij} \cdot \epsilon(t_k - t_m) + C, \tag{9}$$

where $\epsilon(t)$ denotes the spike response kernel (Eq. 3). $C$ denotes the influence of other spikes, which is not in our concern here. In backward pass, according to Eqs. (4)-(5), we have

$$\frac{\partial L}{\partial t_m(s_j)} = \frac{\partial L}{\partial t_k(s_i)} \cdot \frac{\partial t_k(s_i)}{\partial u_i(t_k)} \cdot \frac{\partial u_i(t_k)}{\partial t_m(s_j)} = \frac{\partial L}{\partial t_k(s_i)} \cdot \frac{-1}{du_i(t_k)/dt} \cdot w_{ij} \cdot \frac{\partial \epsilon(t_k - t_m)}{\partial t_m}. \tag{10}$$

Note that when a spike is emitted by neuron $i$ at time $t_k$, the slope of $u_i(t) > 0$ at time $t_k$, which means $\frac{-1}{du_i(t_k)/dt}$ has a negative sign. Considering $\frac{\partial \epsilon(t_k - t_m)}{\partial t_m} = -\frac{d\epsilon(\tau)}{d\tau}$, where $\tau = t_k - t_m$, we get:

$$\text{sign}\left(\frac{\partial L}{\partial t_m(s_j)}\right) = \text{sign}\left(\frac{\partial L}{\partial t_k(s_i)}\right) \cdot \text{sign}\left(w_{ij}\right) \cdot \text{sign}\left(\frac{d\epsilon(\tau)}{d\tau}\right). \tag{11}$$

When $\text{sign}\left(d\epsilon(\tau)/d\tau\right) = -1$, which is the part of the spike response kernel that decreases (see the case at $t_k'$ in Fig. 4), the gradient direction of $t_m$ can be classified into two cases: When $w_{ij} > 0$, $\text{sign}\left(\frac{\partial L}{\partial t_m(s_j)}\right) = -\text{sign}\left(\frac{\partial L}{\partial t_k(s_i)}\right)$, which means the gradient direction is reversed. When $w_{ij} < 0$, $\text{sign}\left(\frac{\partial L}{\partial t_m(s_j)}\right) = \text{sign}\left(\frac{\partial L}{\partial t_k(s_i)}\right)$, which means the gradient direction is kept.

In both cases, the sign of the gradient gets wrong in propagation between layers. Thus, the commonly used double-exponential spike response kernel is incompatible with the time-based gradient in event-driven learning.

**A smoother gradient assigning approach.** Inspired by the above gradient inconsistency as well as the invariance of gradient sum, we propose a new gradient backpropagation approach here. Specifically, we replace the function $\frac{\partial \epsilon(t_k - t_m)}{\partial t_m}$ in Eqs. (4) and (5) with a new function $h(t_k - t_m)$. Therefore, the backpropagation formula between layers turns into:

$$\frac{\partial t_k(s_i)}{\partial t_m(s_j)} = \frac{\partial t_k(s_i)}{\partial u_i(t_k)} \cdot \frac{\partial u_i(t_k)}{\partial t_m(s_j)} \tag{12}$$

$$= \begin{cases} 0, & \text{if} \quad t_m(s_j) \leq t_{k,last}(s_i) \text{ or } t_m(s_j) > t_k(s_i), \\ \left(\sum_{t_{k,last}(s_i) < t_m(s_j') \leq t_k(s_i)} w_{ij} \cdot h(t_k - t_m)\right)^{-1} \cdot w_{ij} \cdot h(t_k - t_m), & \text{otherwise.} \end{cases}$$

It can be see from Eq. 12 that $\frac{\partial t_k(s_i)}{\partial t_m(s_j)}$ will not change if we multiply $h(t)$ by an arbitrary constant, so we do not need to care about the scale of $h(t)$. Meanwhile, the property of invariable sum of gradients is kept after this replacement.

To guarantee that the gradients are not reversed between layers, we should expect $h(t) > 0$ always hold when $t > 0$. Therefore, we choose $h(t) = e^{-\frac{t}{\tau_{grad}}}$ to simplify the calculation, where $\tau_{grad}$ is a tunable parameter. Notice that the function $h(t)$ is only used in backward propagation, which means the spike response kernel in the forward propagation is not necessarily the integral of $h(t)$.

### 3.4 Overall Learning Rule

The loss function we use in this work is the counting loss function, which has the form $L = \frac{1}{N} \sum_{i=1}^{N_{out}} \left(\frac{1}{T}\left(N_i^{target} - \int_0^T s_i(t)dt\right)\right)^2$, where $N_{out}$ is the number of output neurons and equals to the number of classes, $s_i(t)$ represents the spike train emitted by neuron $i$. Besides, $N_i^{target}$ is the target of the spike number outputted by neuron $i$ and typically we set $N_i^{target}$ larger when $i$ is the correct answer.

During the learning process, the gradient is first propagated from the loss function to the firing time of each spike from the last layer to the first layer. The formula for this stage is (please refer to Appendix for the detailed deduction):

$$\frac{\partial L}{\partial t_m(s_j^{(l-1)})} = \sum_i \frac{\partial L}{\partial t_{k,next}(s_i^{(l)})} \cdot \left( \sum_{t_i^{last}(s_i^{(l)}) < t_m(s_j^{(l-1)}) \leq t_{k,next}(s_i^{(l)})} w_{ij}^{(l)} \cdot h(t_{k,next} - t_m) \right)^{-1}$$
$$\cdot w_{ij}^{(l)} \cdot h(t_{k,next} - t_m), \tag{13}$$

where $t_{k,next}(s_i^{(l)})$ is the firing time of the first spike emitted by neuron $i$ after time $t_m$.

After this, the gradient to weights in each layer is calculated by summing up the multiplication of the gradients of spike firing times in the same layer and the derivative of weights with respect to spike firing times. The learning rule for this stage is

$$\frac{\partial L}{\partial w_{ij}^{(l)}} = \sum_{t_m(s_j^{(l-1)})} \frac{\partial L}{\partial t_{k,next}(s_i^{(l)})} \cdot \frac{-1}{\frac{\partial u_i^{(l)}(t_{k,next})}{\partial t}} \cdot \epsilon(t_{k,next} - t_m). \tag{14}$$

## 4 Experiments

In this section, we validate the effectiveness of our method on MNIST [48], Fashion-MNIST [49], N-MNIST [50], CIFAR10 [51], and CIFAR100 [51] datasets. This section is organized as follows: We first introduce the training details, then evaluate the performance of our algorithm and compare it with the state-of-the-art event-driven learning approaches. At last, we conduct ablation studies to illustrate the effectiveness of our proposed modules. More details of the configurations can be found in the Appendix.

### 4.1 Training Details

**Initialization:** When training in an event-driven fashion, gradient information is only carried by spikes. Therefore, the gradient information will be completely blocked by a layer when there are no spikes in that layer. To solve this problem, we start with layers of arbitrarily initialized weights and scale them by certain multiples, which can make the average firing rate to be a certain number for each layer. We obtain these multiple parameters by binary search and this strategy works well in practice.

**Supervisory signals:** Another problem we face is that output neurons corresponding to certain classes do not fire anymore after certain epochs of training. This problems makes corresponding gradients difficult to propagate in the network, further leading to these neurons no longer fire afterwards, resulting recognizing those classes correctly impossible in the following epochs. To address this problem, we utilize supervisory signals. For each neuron in the output layer corresponding to the ground-truth label, we force it to fire at the end of the simulation.

**Experiment settings:** In our experiments, we use the real-valued spike current representing the pixel intensities of the image as inputs. We list the network architecture each work uses and the accuracy they achieves on each dataset in table 2. Notice that the output layer is, by default, a fully-connected layer containing the same number of neurons as the number of classes in the dataset, and omitted from the architecture representation. We run all experiments on a single Nvidia A100 GPU.

### 4.2 Comparison with the State-of-the-Art

Tab. 2 reports the accuracies of the proposed method and other comparing methods. The performance of our algorithm is lower than TSSL-BP by 0.06% on the MNIST dataset and 0.01% on the N-MNIST dataset. However, the output of their network is real-valued postsynaptic currents while the output of our network is binary spikes. In addition, they use RNN-like gradients to assist learning. On the remaining datasets, we have achieved state-of-the-art performance among these works with temporal gradients. For the Fashion-MNIST dataset, our algorithm performs 0.45% higher than the previous SOTA. On the CIFAR10 dataset, we achieve 92.45% accuracy with a 14-layer SEW-Resnet and

Table 2: Performance comparison

| Dataset | Model | Gradient Type | Architecture | Accuracy |
|---|---|---|---|---|
| MNIST | Mostafa [43] | Temporal | 784-800 | 97.5% |
| | Cosma [34] | Temporal | 784-340 | 97.9% |
| | S4NN [35] | Temporal | 784-600 | 97.4% |
| | BS4NN [52] | Temporal | 784-600 | 97.0% |
| | STiDi-BP [36] | Temporal | 784-500 | 97.4% |
| | TSSL-BP [39] | Temporal | 15C5-P2-40C5-P2-300[a] | 99.53% |
| | STDBP [38] | Temporal | 16C5-P2-32C5-P2-800-128 | 99.4% |
| | Ours | Temporal | 15C5-P2-40C5-P2-300 | 99.47% |
| Fashion-MNIST | S4NN [35] | Temporal | 784-1000 | 88.0% |
| | BS4NN [52] | Temporal | 784-1000 | 87.3% |
| | STDBP [38] | Temporal | 16C5-P2-32C5-P2-800-128 | 90.1% |
| | TSSL-BP [39] | Temporal | 32C5-P2-64C5-P2-1024 | 92.83% |
| | Ours | Temporal | 32C5-P2-64C5-P2-1024 | 93.28% |
| N-MNIST | TSSL-BP [39] | Temporal | 12C5-P2-64C5-P2 | 99.40% |
| | Ours | Temporal | 12C5-P2-64C5-P2 | 99.39% |
| CIFAR10 | TSSL-BP [39] | Temporal | CIFARNet[b] | 91.41% |
| | Ours | Temporal | VGG11[c] | 92.10% |
| | Ours | Temporal | SEW-Resnet14 | 92.45% |
| CIFAR100 | Ours | Temporal | VGG11 | 63.97% |

[a] 15C5: convolution layer with 15 channels of $5 \times 5$ filters. P2: pooling layer with $2 \times 2$ filters.
[b] CIFARNet: 128C3-256C3-P2-512C3-P2-1024C3-512C3-1024-512
[c] VGG11: 128C3-128C3-P2-256C3-256C3-256C3-P2-512C3-512C3-512C3-P2-2048-2048

92.10% with VGG11, which are all better than the current SOTA, 91.41%. For the CIFAR100 dataset, we are the first work to successfully train SNNs with time-based gradients in an event-driven fashion. We have achieved a performance of 63.97% on this dataset.

## 4.3 Ablation Studies

To show the effect of our proposed modules, we conduct ablation experiments on the CIFAR10 dataset. Specifically, two proposed components are taken into consideration: (1) As mentioned in Section 3.3, we compare the proposed gradient assignment functions $h(t) = e^{-\beta t}$ in (12) with the commonly used one $h(t) = \frac{d\epsilon(t)}{dt}$. (2) We compare the results of three different types of pooling layers (average pooling, adjusted average pooling and max pooling) mentioned in Section 3.2. We have tried all combinations of gradient assignment functions and pooling layers. The test accuracy of these different settings is shown in Tab. 3.

Table 3: Test accuracy comparison between different settings on CIFAR10

| | Average pooling | Adjusted average pooling | Max pooling |
|---|---|---|---|
| $\frac{d\epsilon(t)}{dt}$ | 89.58% | 91.52% | 91.64% |
| $e^{-\beta t}$ | 89.80% | 91.88% | 91.79% |

The results in Tab. 3 meets our expectation. For the pooling layer, max pooling and adjusted average pooling have much better performance than the average pooling. This accords with the conclusion in Section 3.2 that pooling layers keeping the property of invariant sum of gradients are better than those that do not.

The proposed gradient assignment function $h(t) = e^{-\beta t}$ is also better than the commonly used one $h(t) = \frac{d\epsilon(t)}{dt}$ for all three types of pooling layers. In addition, as shown in Fig. 5, $h(t) = e^{-\beta t}$ converges faster than $h(t) = \frac{d\epsilon(t)}{dt}$ in early stage.

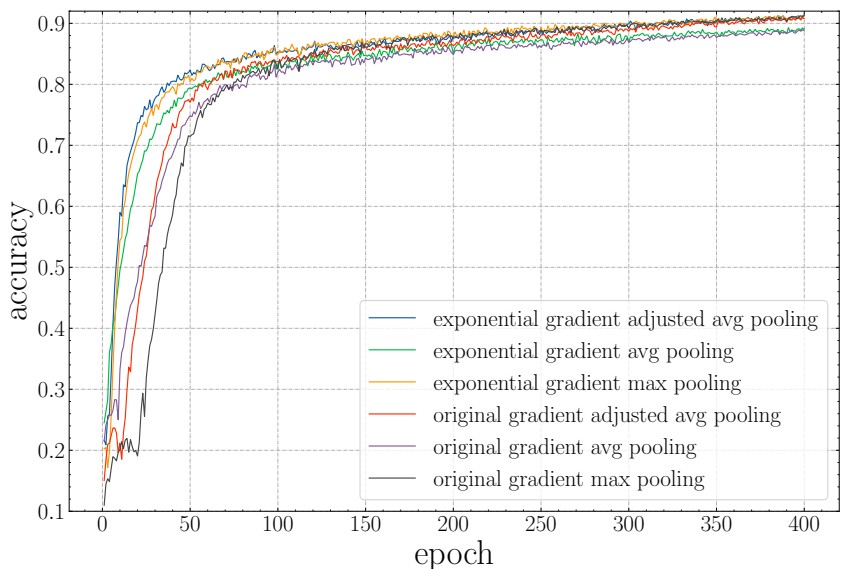

Figure 5: Test accuracy curve comparison between different settings on CIFAR10 (exponential gradient denotes $h(t) = e^{-\beta t}$ and original gradient denotes $h(t) = \frac{d\epsilon(t)}{dt}$).

## 5 Conclusion and Discussion

In this work, we analyze the commonly-used SNN temporal backpropagation training approach and find that it follows the gradient assignment rule. We also find the average pooling layer does not obey this rule while the max pooling layer does. We show that the direction of the temporal gradient will be reversed when the spike kernel is decreasing and avoid it with an increasing kernel in backpropagation. Our algorithm achieves state-of-the-art performance on CIFAR10 among time-based SNN learning approaches and successfully learns the parameters of SNN on CIFAR100 for the first time.

Compared with RNN-like methods, the proposed event-based learning algorithm has a lower computational cost and memory occupation when there are many time steps. Besides, our algorithm also does not need bias between layers. Meanwhile, gradient propagation between spikes instead of time steps can mitigate the gradient explosion/vanishing problem along the time axis.

However, there is still a gap between event-driven backpropagation and biological plausible learning, since event-driven backpropagation processes the spike train in reverse time, which conflicts with the online learning in the real world and desires for future research.

## 6 Acknowledgements

This work was supported by the National Natural Science Foundation of China Grants 62176003 and 62088102.

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
