# Supplementary Material of Training Spiking Neural Networks with Event-driven Backpropagation

## A  Invariant sum of gradients among layers with weights

The most commonly used time-based gradient backpropagation method origins from [1]. It approximates the derivative of spike firing time with respect to membrane potential as the negative inverse of the derivative of membrane potential to time: $\frac{\partial t_k(s_i^{(l)})}{\partial u_i^{(l)}(t_k)} = \frac{-1}{du_i^{(l)}(t_k)/dt}$, where $t_k(s_i^{(l)})$ is the firing time of a spike emitted by neuron $i$ in layer $l$ at time $t_k$ and $u_i^{(l)}(t_k)$ is the membrane potential of neuron $i$ in layer $l$ at time $t_k$.

Generally, we want to calculate derivatives of spike firing times between layers. Denote $t_m(s_j^{(l-1)})$ as the firing time of a spike emitted by neuron $j$ in layer $l-1$ at time $t_m$, we want to get the value $\frac{\partial t_k(s_i^{(l)})}{\partial t_m(s_j^{(l-1)})} = \frac{\partial t_k(s_i^{(l)})}{\partial u_i^{(l)}(t_k)} \cdot \frac{\partial u_i^{(l)}(t_k)}{\partial t_m(s_j^{(l-1)})}$. To obtain the key derivatives $\frac{\partial t_k(s_i^{(l)})}{\partial u_i^{(l)}(t_k)}$ and $\frac{\partial u_i^{(l)}(t_k)}{\partial t_m(s_j^{(l-1)})}$, we first need to rewrite the formula of $u_i^{(l)}(t_k)$:

$$
\begin{aligned}
u_i^{(l)}(t) &= \int_{t_{i,last}^{(l)}}^{t} \left( \sum_j w_{ij}^{(l)} \cdot s_j^{(l-1)}(\tau) \right) \cdot \epsilon(t - \tau) d\tau \\
&= \sum_j \sum_{t_{i,last}^{(l)} < t_m(s_j^{(l-1)}) \leq t} w_{ij}^{(l)} \cdot \epsilon(t - t_m),
\end{aligned}
\tag{S1}
$$

where $t_{i,last}^{(l)}$ is the firing time of the last spike emitted by neuron $i$ in layer $l$ before time $t$, $w_{ij}^{(l)}$ is the connection weight between neuron $j$ in layer $l-1$ and neuron $i$ in layer $l$, $\epsilon()$ is the spike response kernel and $s_j^{(l-1)}(\tau) = \delta(u_j^{(l-1)}(\tau) - \theta)$, where $\delta$ is a Dirac delta function and $\theta$ is the firing threshold. This equation comes from the fact that $s_j^{(l-1)}(\tau)$ is a Dirac delta function at firing time, which means its integration is 1 at the firing time. As a result, we can get:

$$
\begin{aligned}
\frac{\partial t_k(s_i^{(l)})}{\partial u_i^{(l)}(t_k)} = \frac{-1}{du_i^{(l)}(t_k)/dt} &= -\left( \sum_j \sum_{t_{k,last}(s_i^{(l)}) < t_m(s_j^{(l-1)}) \leq t_k(s_i^{(l)})} w_{ij}^{(l)} \cdot \frac{\partial \epsilon(t_k - t_m)}{\partial(t_k - t_m)} \right)^{-1} \\
&= \left( \sum_j \sum_{t_{k,last}(s_i^{(l)}) < t_m(s_j^{(l-1)}) \leq t_k(s_i^{(l)})} w_{ij}^{(l)} \cdot \frac{\partial \epsilon(t_k - t_m)}{\partial t_m} \right)^{-1},
\end{aligned}
\tag{S2}
$$

$$
\frac{\partial u_i^{(l)}(t_k)}{\partial t_m(s_j^{(l-1)})} = w_{ij}^{(l)} \cdot \frac{\partial \epsilon(t_k - t_m)}{\partial t_m},
\tag{S3}
$$

36th Conference on Neural Information Processing Systems (NeurIPS 2022).

where $t_{k,last}(s_i^{(l)})$ denotes the firing time of the last spike emitted by neuron $i$ before time $t_k$.

From these two equations, we can obtain an invariant equality:

$$
\sum_j \sum_{t_{k,last}(s_i^{(l)})<t_m(s_j^{(l-1)})\leq t_k(s_i^{(l)})} \frac{\partial t_k(s_i^{(l)})}{\partial t_m(s_j^{(l-1)})}
$$

$$
= \sum_j \sum_{t_{k,last}(s_i^{(l)})<t_m(s_j^{(l-1)})\leq t_k(s_i^{(l)})} \frac{\partial t_k(s_i^{(l)})}{\partial u_i^{(l)}(t_k)} \cdot \frac{\partial u_i^{(l)}(t_k)}{\partial t_m(s_j^{(l-1)})}
$$

$$
= \frac{\partial t_k(s_i^{(l)})}{\partial u_i^{(l)}(t_k)} \cdot \sum_j \sum_{t_{k,last}(s_i^{(l)})<t_m(s_j^{(l-1)})\leq t_k(s_i^{(l)})} \frac{\partial u_i^{(l)}(t_k)}{\partial t_m(s_j^{(l-1)})}
$$

$$
= \left( \sum_j \sum_{t_{k,last}(s_i^{(l)})<t_m(s_j^{(l-1)})\leq t_k(s_i^{(l)})} w_{ij}^{(l)} \cdot \frac{\partial \epsilon(t_k - t_m)}{\partial t_m} \right)^{-1}
$$

$$
\cdot \sum_j \sum_{t_{k,last}(s_i^{(l)})<t_m(s_j^{(l-1)})\leq t_k(s_i^{(l)})} w_{ij}^{(l)} \cdot \frac{\partial \epsilon(t_k - t_m)}{\partial t_m}
$$

$$
= 1. \tag{S4}
$$

Since the spike firing time is determined by the membrane potential at spike time, which is only influenced by input spikes after the last output spike, the gradients of spike timings between layers are zero out of certain ranges:

$$
\frac{\partial t_k(s_i^{(l)})}{\partial t_m(s_j^{(l-1)})} = 0, \quad \text{if } t_m(s_j^{(l-1)}) \leq t_{k,last}(s_i^{(l)}) \text{ or } t_m(s_j^{(l-1)}) > t_k(s_i^{(l)}). \tag{S5}
$$

Therefore, if we denote the loss function as $L$, then the gradient of $L$ with respect to $t_m(s_j^{(l-1)})$ is:

$$
\frac{\partial L}{\partial t_m(s_j^{(l-1)})} = \sum_i \sum_{t_k(s_i^{(l)})} \frac{\partial L}{\partial t_k(s_i^{(l)})} \cdot \frac{\partial t_k(s_i^{(l)})}{\partial t_m(s_j^{(l-1)})}
$$

$$
= \sum_i \sum_{t_m(s_j^{(l-1)})<t_k(s_i^{(l)})\leq t_{m,next}(s_j^{(l-1)})} \frac{\partial L}{\partial t_k(s_i^{(l)})} \cdot \frac{\partial t_k(s_i^{(l)})}{\partial t_m(s_j^{(l-1)})}, \tag{S6}
$$

where $t_{m,next}(s_j^{(l-1)})$ is the firing time of the next spike emitted by neuron $j$ in layer $l-1$ after time $t_m$.

By summing up all $\frac{\partial L}{\partial t_m(s_j^{(l-1)})}$-s of a layer, we can get another invariant between layers:

$$\sum_j \sum_{t_m(s_j^{(l-1)})} \frac{\partial L}{\partial t_m(s_j^{(l-1)})}$$

$$= \sum_j \sum_{t_m(s_j^{(l-1)})} \sum_i \sum_{t_m(s_j^{(l-1)})<t_k(s_i^{(l)})\leq t_{m,next}(s_j^{(l-1)})} \frac{\partial L}{\partial t_k(s_i^{(l)})} \cdot \frac{\partial t_k(s_i^{(l)})}{\partial t_m(s_j^{(l-1)})}$$

$$= \sum_i \sum_j \sum_{t_m(s_j^{(l-1)})} \sum_{t_m(s_j^{(l-1)})<t_k(s_i^{(l)})\leq t_{m,next}(s_j^{(l-1)})} \frac{\partial L}{\partial t_k(s_i^{(l)})} \cdot \frac{\partial t_k(s_i^{(l)})}{\partial t_m(s_j^{(l-1)})}$$

$$= \sum_i \sum_j \sum_{t_k(s_i^{(l)})} \sum_{t_{k,last}(s_i^{(l)})<t_m(s_j^{(l-1)})\leq t_k(s_i^{(l)})} \frac{\partial L}{\partial t_k(s_i^{(l)})} \cdot \frac{\partial t_k(s_i^{(l)})}{\partial t_m(s_j^{(l-1)})}$$

$$= \sum_i \sum_{t_k(s_i^{(l)})} \frac{\partial L}{\partial t_k(s_i^{(l)})} \cdot \sum_j \sum_{t_{k,last}(s_i^{(l)})<t_m(s_j^{(l-1)})\leq t_k(s_i^{(l)})} \frac{\partial t_k(s_i^{(l)})}{\partial t_m(s_j^{(l-1)})}$$

$$= \sum_i \sum_{t_k(s_i^{(l)})} \frac{\partial L}{\partial t_k(s_i^{(l)})}, \tag{S7}$$

where $t_{m,next}(s_j^{(l-1)})$ is the first spike fired by neuron $j$ in layer $l-1$ after time $t_m$. Among these equations, the third line uses the fact that $t_k(s_i^{(l)})$ being between $t_m(s_j^{(l-1)})$ and its next spike is equivalent to $t_m(s_j^{(l-1)})$ being between $t_k(s_i^{(l)})$ and its last spike. The last line uses formula (S4).

## B    Deduction of backward formulas

In this section, we calculate all gradients of weights with respect to the loss, which can be represented as

$$\frac{\partial L}{\partial w_{ij}^{(l)}} = \sum_{t_k(s_i^{(l)})} \frac{\partial L}{\partial t_k(s_i^{(l)})} \cdot \frac{\partial t_k(s_i^{(l)})}{\partial w_{ij}^{(l)}}. \tag{S8}$$

As a result, we have to first calculate the gradients of spike timings to the loss.

Firstly we have to rewrite the expression of the loss:

$$L = \frac{1}{N} \sum_{i=1}^{N_{out}} \left( \frac{1}{T} \left( N_i^{target} - \int_0^T s_i^{(n)}(t)dt \right) \right)^2$$

$$= \frac{1}{N} \sum_{i=1}^{N_{out}} \left( \frac{1}{T} \left( N_i^{target} - \sum_{t_k(s_i^{(n)})} 1 \right) \right)^2, \tag{S9}$$

where $N_i^{target}$ is the target number of spikes fired by neuron $i$ in the output layer.

For the output layer, we make an approximation that $\partial(\sum_{t_k(s_i^{(n)})} 1)/\partial t_k = -1$. This approximation is based on the fact that making $t_k$ bigger results in smaller weights, which pushes $\partial(\sum_{t_k(s_i^{(n)})} 1)/\partial t_k$ smaller.

Therefore, the derivative of spike firing time with respect to the loss is

$$\frac{\partial L}{\partial t_k(s_i^{(n)})} = \frac{2}{NT^2} \left( N_i^{target} - \sum_{t_k(s_i^{(n)})} 1 \right). \tag{S10}$$

for each neuron $i$ and spike time $t_k$, where the spiking neural network has $n$ layers in total.

For previous layers, one input spike of a neuron only influences the first output spike after it of that neuron directly. Here we apply a smoother gradient assignment method according to section 3.3 in the main text, which replaces $\frac{\partial \epsilon(t_k - t_m)}{\partial t_m}$ with $h(t_k - t_m)$ in equations (S2) (S3). Then we can deduce the derivation of the loss to a certain spike firing time:

$$
\begin{aligned}
\frac{\partial L}{\partial t_m(s_j^{(l-1)})} &= \sum_i \frac{\partial L}{\partial t_{k,next}(s_i^{(l)})} \cdot \frac{\partial t_{k,next}(s_i^{(l)})}{\partial t_m(s_j^{(l-1)})} \\
&= \sum_i \frac{\partial L}{\partial t_{k,next}(s_i^{(l)})} \cdot \frac{\partial t_{k,next}(s_i^{(l)})}{\partial u_i^{(l)}(t_{k,next})} \cdot \frac{\partial u_i^{(l)}(t_{k,next})}{\partial t_m(s_j^{(l-1)})} \\
&= \sum_i \frac{\partial L}{\partial t_{k,next}(s_i^{(l)})} \cdot \left( \sum_{t_{k,last}(s_i^{(l)}) < t_{m_1}(s_j^{(l-1)}) \leq t_{k,next}(s_i^{(l)})} w_{ij}^{(l)} \cdot h(t_{k,next} - t_{m_1}) \right)^{-1} \\
&\quad \cdot w_{ij}^{(l)} \cdot h(t_{k,next} - t_m),
\end{aligned}
\tag{S11}
$$

where $t_{k,next}(s_i^{(l)})$ is the timing of the next spike fired by neuron $i$ in layer $l$ after the spike $s_j^{(l-1)}(t_m)$, $t_{k,last}(s_i^{(l)})$ is the timing of the last spike fired by neuron $i$ in layer $l$ before the spike $s_j^{(l-1)}(t_m)$, and $u_i^{(l)}(t_{k,next})$ is the membrane potential of neuron $i$ in layer $l$ at time $t_{k,next}(s_i^{(l)})$.

Then we return to the gradients of weights. Here we take the approximations same as [1]:

$$
\begin{aligned}
\frac{\partial t_k(s_i^{(l)})}{\partial w_{ij}^{(l)}} &= \frac{\partial t_k(s_i^{(l)})}{\partial u_i^{(l)}(t_k)} \cdot \frac{\partial u_i^{(l)}(t_k)}{\partial w_{ij}^{(l)}} \\
&= \frac{-1}{\partial u_i^{(l)}(t_k)/\partial t} \cdot \frac{\partial \sum_{j1} \sum_{t_{k,last}(s_i^{(l)}) < t_m(s_{j1}^{(l-1)}) \leq t_k(s_i^{(l)})} w_{i,j1}^{(l)} \cdot \epsilon(t_k - t_m)}{\partial w_{ij}^{(l)}} \\
&= \frac{-1}{\partial u_i^{(l)}(t_k)/\partial t} \cdot \sum_{t_{k,last}(s_i^{(l)}) < t_m(s_j^{(l-1)}) \leq t_k(s_i^{(l)})} \epsilon(t_k - t_m).
\end{aligned}
\tag{S12}
$$

In the second line, we use equation (S1) to replace $u_i^{(l)}(t_k)$.

Combining formula (S11) (S12), we have

$$
\begin{aligned}
\frac{\partial L}{\partial w_{ij}^{(l)}} &= \sum_{t_k(s_i^{(l)})} \frac{\partial L}{\partial t_k(s_i^{(l)})} \cdot \frac{\partial t_k(s_i^{(l)})}{\partial u_i^{(l)}(t_k)} \cdot \frac{\partial u_i^{(l)}(t_k)}{\partial w_{ij}^{(l)}} \\
&= \sum_{t_k(s_i^{(l)})} \frac{\partial L}{\partial t_k(s_i^{(l)})} \cdot \frac{-1}{\frac{\partial u_i^{(l)}(t_k)}{\partial t}} \cdot \sum_{t_{k,last}(s_i^{(l)}) < t_m(s_j^{(l-1)}) \leq t_k(s_i^{(l)})} \epsilon(t_k - t_m) \\
&= \sum_{t_k(s_i^{(l)})} \sum_{t_{k,last}(s_i^{(l)}) < t_m(s_j^{(l-1)}) \leq t_k(s_i^{(l)})} \frac{\partial L}{\partial t_k(s_i^{(l)})} \cdot \frac{-1}{\frac{\partial u_i^{(l)}(t_k)}{\partial t}} \cdot \epsilon(t_k - t_m) \\
&= \sum_{t_m(s_j^{(l-1)})} \sum_{t_k = t_{k,next}(s_i^{(l)})} \frac{\partial L}{\partial t_k(s_i^{(l)})} \cdot \frac{-1}{\frac{\partial u_i^{(l)}(t_k)}{\partial t}} \cdot \epsilon(t_k - t_m) \\
&= \sum_{t_m(s_j^{(l-1)})} \frac{\partial L}{\partial t_{k,next}(s_i^{(l)})} \cdot \frac{-1}{\frac{\partial u_i^{(l)}(t_{k,next})}{\partial t}} \cdot \epsilon(t_{k,next} - t_m).
\end{aligned}
\tag{S13}
$$

In summary, equations (S13) and (S11) constitutes the gradient backpropagation formulas in the continuous form.

## C  Time complexity analysis

In this section, we show that the number of operations of the event-driven learning algorithms is less than the RNN-based learning algorithms when spikes are sparse. For simplicity, we only analyze a single fully-connected layer with M input neurons and N output neurons. Other layers and the whole network can be analyzed in a similar way.

During training, RNN-based learning algorithms are forced to unfold through the time axis, as explained in Figure 1 and Section 2. As a result, the corresponding number of operations is at least $O(TMN)$, where T is the total time steps and M, N are the number of input and output neurons. On the other hand, event-based learning algorithms only have to deal with cases where a certain neuron fires a spike, and record the relevant information. In the forward stage, a spike fired by an input neuron affects the state itself and all output neurons, which is $O(N)$ in total. In the backward stage, a spike fired by an output neuron needs to propagate gradient information to all spikes between this spike and the last spike fired by this neuron (so all input spikes are processed once in this stage). Therefore, denoting the average firing rate of input and output neurons to be $\alpha$ and $\beta$, the number of operations of this layer is $O(T(\alpha MN + \beta M + \alpha N)) = O(T(\alpha MN + \beta M))$. When spikes are sparse, event-based learning algorithms certainly have advantages since $\alpha + \beta \ll 1$ in this case.

## D  Discrete simulation

Although event-driven learning should be simulated in continuous time by nature, existing deep learning frameworks are not suitable for continuous-time simulation when one neuron could fire more than one spike. As a result, we choose to simulate event-driven learning in discrete time steps. In the discrete-time simulation, some approximation should be made.

There are two major differences between the discrete form and the continuous form: The first is that in the discrete form, spikes are forced to emit at integer time steps, where some errors might occur. The second is that the structured spike emitting and the status update in discrete form makes the formulas easily represented in vector and matrix form.

In the following part, we will deduce forward formulas and backward formulas in discrete form in detail.

**Forward formulas:**

$$u^{(l)}[t] = (\epsilon * ((\boldsymbol{W}^{(l)} \cdot s^{(l-1)})[t_{last}^{(l)} + 1 : t]))[t], \tag{S14}$$

$$s^{(l)}[t] = H(u^{(l)}[t] - \theta), \tag{S15}$$

where $u^{(l)}[t]$ is the membrane potential (before resetting) vector of all neurons in layer $l$ at time $t$, $t_{last}^{(l)}$ is the vector of last spike time before time $t$ of neurons in layer $l$, and $s^{(l)}[t]$ represents the spikes emitted from neurons layer $l$ at time $t$ (which is 1 when a spike is emitted and 0 otherwise). The length of $u^{(l)}[t], s^{(l)}[t], t_{last}^{(l)}$ are all $N_l$ (which is the number of neurons in layer $l$). The weight matrix $\boldsymbol{W}^{(l)}$ has a size of $N_l \times N_{l-1}$. Notice that $[t_{last}^{(l)} + 1 : t]$ in S14 contains both boundary of the interval.

To simulate for fewer steps, we make a small modification to the kernel $\epsilon()$ into

$$\epsilon(t) = \frac{\tau_m}{\tau_m - \tau_s}(e^{-\frac{t+1}{\tau_m}} - e^{-\frac{t+1}{\tau_s}}), \tag{S16}$$

which makes $\epsilon(t) > 0$ when $t \geq 0$.

**Backward formulas:**

For the gradients of spike timings with respect to the loss in the output layer, we just have to turn equation (S10) into the discrete form

$$\frac{\partial L}{\partial t_k(s_i^{(n)})} = \frac{2}{NT^2}\left(N_i^{target} - \sum_{t_k=1}^{T} s_i^{(n)}[t_k]\right). \tag{S17}$$

For the rest layers, we can first turn (S11) (S8) into discrete form

$$\frac{\partial L}{\partial t_m[s_j^{(l-1)}]} = \sum_i \frac{\partial L}{\partial t_{k,next}[s_i^{(l)}]} \cdot \left( \sum_{t_{i,last}[s_i^{(l)}]<t_{m_1}[s_j^{(l-1)}]\leq t_{k,next}[s_i^{(l)}]} w_{ij}^{(l)} \cdot h(t_{k,next} - t_{m_1}) \right)^{-1}$$
$$\cdot w_{ij}^{(l)} \cdot h(t_{k,next} - t_m), \tag{S18}$$

$$\frac{\partial L}{\partial w_{ij}^{(l)}} = \sum_{t_m=1}^{T} t_m[s_j^{(l-1)}] \cdot \frac{\partial L}{\partial t_{k,next}[s_i^{(l)}]} \cdot \frac{-1}{\frac{\partial u_i^{(l)}[t_{k,next}]}{\partial t}} \cdot \epsilon(t_{k,next} - t_m). \tag{S19}$$

Since $t_m$ and $t_{k,next}$ are integers here, the definition of $t_{k,next}$ is the smallest number when neuron $i$ in layer $l$ emits a spike satisfying $t_{k,next} \geq t_m$. We use $\geq$ instead of $>$ here since when $t_{k,next} = t_m$, spike at time $t_m$ of neuron $j$ in layer $l-1$ can contribute to spike $t_{k,next}$ of neuron $i$ in layer $l$.

It should be noticed that for the same spike $s_j[t_m]$, different post-synaptic neuron $i$ can have different times of next spike $t_{k,next}$-s. Hence we denote $t_{next}^{(l)}$ as a vector of times $t_{k,next}$ for all neuron $i$ in layer $l$. In addition, we denote $g(t_k[s_i^{(l)}]) = \sum_{t_{k,last}[s_i^{(l)}]<t_m[s_j^{(l-1)}]\leq t_k[s_i^{(l)}]} w_{ij}^{(l)} \cdot h(t_k - t_m)$. Then we can turn (S18) (S19) into the vector form

$$\frac{\partial L}{\partial t_m[s^{(l-1)}]} = \boldsymbol{W}^{(l)} \cdot \left( h(t_k - t_m) \odot \frac{1}{g(t_k)} \odot \left. \frac{\partial L}{\partial s^{(l)}[t_k]} \right|_{t_k = t_{next}^{(l)}} \right), \tag{S20}$$

$$\frac{\partial L}{\partial \boldsymbol{W}^{(l)}} = \left( \sum_{t_m=1}^{T} s^{(l-1)}[t_m] \cdot \left( \epsilon(t_k - t_m) \odot \frac{\partial L}{\partial s^{(l)}[t_k]} \odot \left. \frac{-1}{\frac{\partial u^{(l)}[t_k]}{\partial t_k}} \right|_{t_k = t_{next}^{(l)}} \right)^{T} \right)^{T}. \tag{S21}$$

In the equations above, $t_{next}^{(l)}$ means the time of the last spike of neurons in layer $l$ before time $t_k$. Meanwhile, $\odot$ represents the element-wise multiplication.

## E   Normalization on weights

Normalizations are often used to stabilize training in both ANNs [2] and SNNs [3]. However, normalization for spiking neural networks in previous works is often applied to the input current of neurons in discrete time-step, which might involve non-spikes (time-steps with no inputs) in the calculation. This property violates the spirit of event-driven learning, which inspires us to explore a new method to stabilize training.

In this work, we use normalization on weights to stabilize the training of our network. Specifically, we use parameters $W_0, \gamma, \beta$ (correspond to normalized weight, scale, and shift correspondingly) to represent the actual weight $W$ in each layer. We normalize each channel separately, which accords with the original BatchNorm [2]. Denote the normalized weight, scale and shift in the $k$- channel to be $W_0^{(k)}, \gamma^{(k)}$, and $\beta^{(k)}$, then we calculate the weight $W^{(k)}$ according to the following equations:

$$\hat{W}_0^{(k)} = \frac{W_0^{(k)} - E[W_0^{(k)}]}{\sqrt{Var[W_0^{(k)}] + \varepsilon}}, \quad W^{(k)} = \gamma^{(k)} \cdot \hat{W}_0^{(k)} + \beta^{(k)}. \tag{S22}$$

In backward propagation, we calculate the gradients normally according to the chain rule, similar to [2]:

$$\frac{\partial L}{\partial \hat{x}_i} = \frac{\partial L}{\partial y_i} \cdot \gamma^{(k)}, \tag{S23}$$

$$\frac{\partial L}{\partial \mu_{(k)}} = \sum_{i=1}^{m} \frac{\partial L}{\partial \hat{x}_i} \cdot \frac{-1}{\sqrt{\sigma_{(k)}^2 + \varepsilon}}, \tag{S24}$$

$$\frac{\partial L}{\partial \sigma_{(k)}^2} = \sum_{i=1}^{m} \frac{\partial L}{\partial \hat{x}_i} \cdot \left( \hat{x}_i - \mu_{(k)} \right) \cdot \frac{-1}{2} \left( \sigma_{(k)}^2 + \varepsilon \right)^{-3/2}, \tag{S25}$$

$$\frac{\partial L}{\partial x_i} = \frac{\partial L}{\partial \hat{x}_i} \cdot \frac{1}{\sqrt{\sigma_{(k)}^2 + \varepsilon}} + \frac{\partial L}{\partial \mu_{(k)}} \cdot \frac{1}{m} + \frac{\partial L}{\partial \sigma_{(k)}^2} \cdot \frac{2(x_i - \mu_{(k)})}{m}, \tag{S26}$$

$$\frac{\partial L}{\partial \gamma^{(k)}} = \sum_{i=1}^{m} \frac{\partial L}{\partial y_i} \cdot \hat{x}_i, \tag{S27}$$

$$\frac{\partial L}{\partial \beta^{(k)}} = \sum_{i=1}^{m} \frac{\partial L}{\partial y_i}, \tag{S28}$$

where $W^{(k)}$ has $m$ elements in total (which equals to the number of neurons in one channel), $x_i$, $\hat{x}_i$, $y_i$ are the $i$-th element of $W_0^{(k)}$, $\hat{W}_0^{(k)}$, $W^{(k)}$ respectively. In the meantime, $\mu_{(k)} = E[W_0^{(k)}]$ and $\sigma_{(k)}^2 = Var[W_0^{(k)}]$.

## F   Input Encoding

In this work, we directly use the real image pixel values as the input of our network, since the image input is commonly used [4, 5, 6], and encoding methods like Poisson encoding often impede the performance of SNNs. Meanwhile, not applying any encoding will not decrease the efficiency of a network for the following reasons [7]. First, in computer vision, the input representation typically has much fewer channels (e.g., Red, Green, and Blue) than internal representations (e.g., 512). As a result, the first layer of a ConvNet is often the smallest convolution layer, both in terms of parameters and computations [8]. Second, it is relatively easy to handle continuous-valued inputs as fixed-point numbers with $m$ bits of precision. We also try the time-to-first-spike encoding (which is used by [9]), which achieves 90.33% testing accuracy on the CIFAR10 dataset.

## G   Implementation Details

We have listed the architecture trained on each dataset in the main text. We use the cosine-annealing learning rate in training, and add a weight decay of 0.0005 for all datasets. Besides, we clip the L2-norm of the gradient to 1 when training. In initialization, we initialize models for each dataset with different average numbers of spikes per neuron. The hyper-parameters we have used are listed in Table S1. In this table, $T$ is the number of time steps, $\tau_m$ and $\tau_s$ are parameters of the forward kernel in equation (S16), $\tau_{grad}$ is the parameter of the backward kernel $h(t) = e^{-\frac{t}{\tau_{grad}}}$. $N_{correct}^{target}$ and $N_{wrong}^{target}$ are the target number of spikes for output neurons corresponding to the answer and those does not correspond to the answer respectively.

## H   Societal Impact and Limitations

As our work is about training SNN in an event-driven fashion, there is no clear negative social impact. Our proposed method trains large-scale SNN in a more biologically plausible way (than RNN-like methods), which has a more positive social impact. Regarding limitations, our method has not achieved comparable performance to RNN-like approaches. In addition, the event-driven method has not supported instant synapses since the time derivative is infinity for such a case. It should also be noticed that there is still a gap between event-driven backpropagation and biological plausible

Table S1: Values of Hyper-parameters

| Dataset | Optimizer | Learning Rate | T | Initilization Spike Number |
|---------|-----------|---------------|---|----------------------------|
| MNIST | AdamW | 0.0005 | 5 | 0.5 |
| Fashion-MNIST | AdamW | 0.0005 | 5 | 0.5 |
| N-MNIST | AdamW | 0.0005 | 30 | 2 |
| CIFAR10 | SGD | 0.05 | 12 | 1 |
| CIFAR100 | SGD | 0.06 | 16 | 1 |

| Dataset | $\tau_m$ | $\tau_s$ | $\tau_{grad}$ | $N_{correct}^{target}$ | $N_{wrong}^{target}$ |
|---------|----------|----------|---------------|------------------------|----------------------|
| MNIST | 5 | 3 | 2.5 | 5 | 1 |
| Fashion-MNIST | 5 | 3 | 2.5 | 5 | 1 |
| N-MNIST | 8 | 4 | 3 | 15 | 2 |
| CIFAR10 | 7 | 4 | 3.5 | 10 | 1 |
| CIFAR100 | 10 | 6 | 5.5 | 15 | 1 |

learning, since event-driven backpropagation processes the spike train in reverse time, which conflicts with online learning in the real world. These topics desire further research.