# OpenReview forum: "Training Spiking Neural Networks with Event-driven Backpropagation"
_NeurIPS.cc/2022/Conference — NeurIPS 2022 Accept_

### Official Review · Reviewer_tMmb · 2022-07-04

**Rating:** 4
**Confidence:** 3
**Soundness:** 2 fair
**Presentation:** 2 fair
**Contribution:** 2 fair

**Summary:**

The focus of this paper is training of spiking neural networks. Specifically, the paper proposes back-propagation with respect to spike timing. They analyze previous methods and propose a small increment over the current state-of-the-art. The results are not clearly discussed.

**Questions:**

Please refer to Strengths and Weakness for the points.

**Limitations:**

There are no potential negative societal impacts. One major limitation of this work is applicability to neuromorphic hardware and how will the work shown on GPU translate to neuromorphic cores.

**Strengths And Weaknesses:**

Strengths:
- The work focuses on an aspect of the learning algorithms that requires optimization and innovation.

Weakness
1.	It is hard to understand what the axes are for Figure 1.
2.	It is unclear what the major contributions of the paper are. Analyzing previous work does not constitute as a contribution.
3.	It is unclear how the proposed method enables better results. For instance, Table 1 reports similar accuracies for this work compared to the previous ones.
4.	The authors talk about advantages over the previous work in terms of efficiency however the paper does not report any metric that shows it is more efficient to train with this proposed method.
5.	Does the proposed method converge faster compared to previous algorithms?
6.	How does the proposed methods compare against surrogate gradient techniques?
7.	The paper does not discuss how the datasets are converted to spike domain.

---

> ### Author Response · Authors · 2022-08-02
> **Response to Reviewer tMmb (Part 1)**
>
> Thank you for your detailed comments and suggestions for improvement. We would like to address your concerns and answer your questions as follows.
>
> > It is hard to understand what the axes are for Figure 1.
>
> Thanks for pointing it out. The horizontal axis denotes the time for all four sub-figures. The vertical axis is membrane potential in (a)(b) and the four vertical layers in \(c\)(d) are (bottom to top) input spikes, input current, membrane potential, and output spike for a neuron. Through Figure 1, we want to emphasize the difference between event-driven learning and RNN-like learning. Information at the firing time of a spike is enough to conduct forward and backward propagation in event-driven learning. In contrast, information at every time step is needed in RNN-like learning. The key reason is shown in Figure 1 \(c\)(d): RNN-like learning requires gradient propagation (spike $\rightarrow$ membrane potential) in each time step through a surrogate function, whereas event-driven learning does not need this.
> This key difference leads to the fact that event-driven learning has the potential to not rely on the concept of ‘time step’, but to infer and learn in a continuous time.
> We have annotated the axes of Figure 1 \(c\)(d) with time steps and input/output.
>
> > It is unclear what the major contributions of the paper are. Analyzing previous work does not constitute a contribution.
>
> We would like to emphasize that our work focuses on event-driven learning (with temporal gradient) of SNNs. Unlike RNN-like BPTT learning approaches (with activation-based gradient), the gradient information is only carried by spikes instead of both spikes and non-spikes at each time step (shown in Figure 1). This sparse gradient propagation path makes it harder than RNN-like BPTT approaches to train SNNs.
>
> The main contributions of our paper are summarized as follows:
> 1. We prove that the typical SNN temporal backpropagation training approach assigns the gradient of an output spike of a neuron to the input spikes generating it. After summing this assignment rule altogether, we find that the sum of gradients is kept unchanged between layers.
> 2. We analyze the case of the pooling layer (which does not have weights) and find that average pooling does not keep the gradient sum unchanged, but we can modify its backward formulas to meet the requirement. Meanwhile, the max-pooling layer satisfies the rule initially.
> 3. We point out the reverse gradient problem in event-driven learning that the direction of the temporal gradient is reversed during backpropagation when the kernel function of an input spike is decreasing. Then we propose a backward kernel function that addresses this problem while keeping the sum of gradients unchanged between layers.
> 4. The adjusted average pooling layer and the non-decreasing backward kernel enhances the performance of our model as well as the convergence speed. To our best knowledge, our proposed approach achieves state-of-the-art performance on CIFAR10 among event-driven training methods (with temporal gradients) for SNNs. Meanwhile, our method is the first event-driven backpropagation approach that successfully trains SNN on the larger-scale CIFAR100 dataset.
>
> The reason we analyze previous work in section 2 is to introduce the background and related works and make our method easier to understand. We hope this will increase your recognition of our work.
>
> > It is unclear how the proposed method enables better results. For instance, Table 1 reports similar accuracies for this work compared to the previous ones.
>
> Table 1 shows that our method can achieve state-of-the-art performance on Fashion-MNIST and CIFAR-10 datasets. The main contender is the TSSL-BP method, which gets 0.06% higher accuracy on the MNIST dataset. We would like to clarify that TSSL-BP uses surrogate gradients in the backward propagation to support training (see lines 55-56), so we are the first pure event-driven method to achieve these results. In addition, our approach is the first event-driven one to successfully train SNNs on CIFAR100.

---

> > ### Author Response · Authors · 2022-08-02
> > **Response to Reviewer tMmb (Part 2)**
> >
> > > The authors talk about advantages over the previous work in terms of efficiency however the paper does not report any metric that shows it is more efficient to train with this proposed method.
> >
> > To illustrate the advantage of efficiency, we show that the number of operations of the event-based learning algorithms is less than the RNN-based learning algorithms when spikes are sparse. For simplicity, we only analyze a single fully-connected layer with M input neurons and N output neurons. Other layers and the whole network can be analyzed similarly.
> >
> > During training, RNN-based learning algorithms are forced to unfold through the time axis, as explained in Figure 1 and Section 2. As a result, the corresponding number of operations is at least $O(TMN)$, where T is the total time steps and M, N are the number of input and output neurons. On the other hand, event-based learning algorithms only have to deal with cases where a certain neuron fires a spike, and record the relevant information. In the forward stage, a spike fired by an input neuron affects the state itself and all output neurons, which is $O(N)$ in total. In the backward stage, a spike fired by an output neuron needs to propagate gradient information to all spikes between this spike and the last spike fired by this neuron (so all input spikes are processed once in this stage). Therefore, denoting the average firing rate of input and output neurons to be $\alpha$ and $\beta$, the number of operations of this layer is $O(T(\alpha MN+\beta M+\alpha N))=O(T(\alpha MN+\beta M))$. When spikes are sparse, event-based learning algorithms certainly have advantages since $\alpha+\beta \ll 1$ in this case.
> > We have added the analysis in the appendix of the revised paper.
> >
> > > How do the proposed methods compare against surrogate gradient techniques?
> >
> > As illustrated in Section 2 "Backgrounds and Related Work", our proposed method differs from surrogate gradient techniques in two aspects.
> > Firstly, we calculate gradients of spike timings with respect to the loss, while surrogate gradient methods calculate gradients of the 'spike scale' (illustrated in Figure 1d).
> > Secondly, our method is event-driven, which means information propagates only through spikes in both forward and backward propagation. In opposite, surrogate gradient techniques propagate gradient information even when spikes are not emitted (recall that the surrogate gradient approximates $\frac{\partial s}{\partial u}$ whether there is a spike or not).
> >
> > This event-driven property makes our method harder to train compared with the surrogate gradient techniques due to the sparse gradient propagation path. Besides, our learning scheme is relatively new compared with surrogate gradient methods. Thus, the performance and convergence speed of this learning scheme has not yet surpassed the surrogate gradient method currently. However, the event-driven property empowers our learning scheme to be more biologically plausible and have more potential for efficiency optimization when running on neuromorphic hardware.
> >
> > > Does the proposed method converge faster compared to previous algorithms?
> >
> > Since our proposed method aims at a topic with less research, it cannot converge faster than the surrogate-gradient-based methods yet. As discussed above, the gradient information can only be passed through spikes, which is sparser than RNN-like surrogate gradient methods. This leads to difficulty in event-driven training. In addition, event-driven learning is a topic with far less existing research than RNN-like training. Therefore, future research might focus on improving the convergence speed.
> > However, the proposed method fits better into the event-driven nature of SNNs, which makes it more power-friendly when training on neuromorphic hardware and more biologically plausible.

---

> > > ### Author Response · Authors · 2022-08-02
> > > **Response to Reviewer tMmb (Part 3)**
> > >
> > > > The paper does not discuss how the datasets are converted to spike domain.
> > >
> > > Thanks for pointing it out. We directly use the real image pixel values as the input of our network, since the image input is commonly used [1][2][3][4], and encoding methods like Poisson encoding often impede the performance of SNNs.
> > > Meanwhile, directly using images as the input will not decrease the efficiency of a network for the following reasons [5]. First, in computer vision, the input representation typically has much fewer channels (e.g., Red, Green, and Blue) than internal representations (e.g., 512). As a result, the first layer of a ConvNet is often the smallest convolution layer, both in terms of parameters and computations (Szegedy et al., 2014). Second, it is relatively easy to handle continuous-valued inputs as fixed-point numbers with $m$ bits of precision.
> > >
> > > We have also tried the time-to-first-spike encoding (used by [6]), which turns the pixel intensity to the spike firing time of a neuron (higher pixel intensity corresponds to an earlier firing time). We achieved 90.33\% testing accuracy for this encoding with a SEW-ResNet 14 network on the CIFAR10 dataset.
> > > We have added the discussion in the appendix of the revised paper.
> > >
> > > >
> > >
> > >     [1] Zhang, W., & Li, P. (2020). Temporal spike sequence learning via backpropagation for deep spiking neural networks. Advances in Neural Information Processing Systems, 33, 12022-12033.
> > >     [2] Kim, Y., Park, H., Moitra, A., Bhattacharjee, A., Venkatesha, Y., & Panda, P. (2022). Rate Coding Or Direct Coding: Which One Is Better For Accurate, Robust, And Energy-Efficient Spiking Neural Networks?. In ICASSP 2022-2022 IEEE International Conference on Acoustics, Speech and Signal Processing (ICASSP) (pp. 71-75). IEEE.
> > >     [3] Kim, Y., Li, Y., Park, H., Venkatesha, Y., & Panda, P. (2022). Neural architecture search for spiking neural networks. arXiv preprint arXiv:2201.10355.
> > >     [4] Wu, Y., Deng, L., Li, G., Zhu, J., Xie, Y., & Shi, L. (2019). Direct training for spiking neural networks: Faster, larger, better. In Proceedings of the AAAI Conference on Artificial Intelligence (Vol. 33, No. 01, pp. 1311-1318).
> > >     [5] Courbariaux, M., Hubara, I., Soudry, D., El-Yaniv, R., & Bengio, Y. (2016). Binarized neural networks: Training deep neural networks with weights and activations constrained to+ 1 or-1. arXiv preprint arXiv:1602.02830.
> > >     [6] Zhang, M., Wang, J., Wu, J., Belatreche, A., Amornpaisannon, B., Zhang, Z., ... & Li, H. (2021). Rectified linear postsynaptic potential function for backpropagation in deep spiking neural networks. IEEE Transactions on Neural Networks and Learning Systems, 33(5), 1947-1958.

---

> ### Author Response · Authors · 2022-08-06
> **Thank you for the time and we hope our response helps to address your questions.**
>
> Thanks for your thorough initial comments. We would like to know whether our response has addressed your questions appropriately. As the discussion period will end soon, we sincerely hope to receive your further feedback, and we are glad to provide a follow-up response if needed.

---

> ### Author Response · Authors · 2022-08-08
> **Thank you for the time and look forward to your feedback**
>
> Dear Reviewer tMmb,
>
> We notice that your initial review can be grouped into two overall concerns and two detailed concerns.
> 1) The main concern is about the contribution of our paper and how our method leads to better results. We would like to note that our time-based event-driven training approach **is essentially different from activation-based surrogate gradient methods and has not been intensively researched (especially on large datasets like CIFAR100)**. The contribution part has been re-summarized in the response below, as well as how the proposed components enhance the performance.
> 2) The secondary concern is the comparison between our method and the surrogate gradient method, including its performance, rate of convergence, and efficiency. We have analyzed the advantage of time complexity of our approach (which received positive feedback from Reviewer kF1o) and explained why the time-based event-driven approach has not yet reached comparable performance and rate of convergence in the following response.
> 3) The third question is about Figure 1, which we have explained in our response below.
> 4) The fourth question is about the input encoding, which we have provided the encoding we use in the response as well as an experiment on the TTFS (time-to-first-spike) encoding (90.33\% accuracy on SEW-Resnet 14 with TTFS encoding compared with 92.45\% accuracy on SEW-Resnet 14 with the original encoding in reply to Reviewer kF1o).
>
> Based on these facts and positive feedback from other reviewers, we sincerely hope you could re-consider your initial rating. If you still have any further comments or questions, please let us know and we are glad to address your further concerns.

---

### Official Review · Reviewer_neur · 2022-07-11

**Rating:** 8
**Confidence:** 2
**Soundness:** 4 excellent
**Presentation:** 4 excellent
**Contribution:** 3 good

**Summary:**

The authors propose a modified event-driven backpropagation and investigate its performance on benchmarks. The authors also investigate if the backpropagation followed a gradient assignment rule, finding that max-pooling obeyed this rule. This is one of the most important conclusions of the paper. The algorithm achieved SOTA on CIFAR-10, and was the first to be trained on CIFAR-100.

**Questions:**

In line 233 you stated that more details about the configuration of training were available on appendix, I was not able to fin the appendix section, nor a appendix file on the supplement material, even though the code contains the configurations of the experiment. It is not clear for me if you want to provide more information in an appendix section or you were referring to the code provided.

**Limitations:**

Spiking Neural Networks has yet achieved SOTA in no benchmark.
Although spiking neural networks are believed to have greater biological plausibility, it is not clear whether biological neural networks learn through backpropagation, which was the method tested in this study. Despite this, nowadays there is no alternative that works better than backpropagation.
The authors also stated that another gap in biological plausibility is the reverse time processing feature.

**Strengths And Weaknesses:**

Spiking neural networks represent a new paradigm of neural networks that, among other advantages, incorporates time into the building blocks of its own functioning. Thus, in addition to having greater biological plausibility, it is also believed to be more coherent with learning in the real-world, which contains the time dimension.
Event-base learning paradigm preserves de biological plausibility and the event-based features advantages of spiking neural networks when compared with surogate backpropagation.
However, this new paradigm has not yet reached SOTA in any benchmark, not even in those that demand incorporation of the time dimension.

---

> ### Author Response · Authors · 2022-08-02
> **Response to Reviewer neur**
>
> Thank you for your positive and constructive feedback. We are encouraged that you find our learning paradigm preserves de biological plausibility and the event-based features advantages of spiking neural networks when compared with surrogate backpropagation. We would like to address your concerns and answer your questions in the following.
>
> > Spiking Neural Networks has yet achieved SOTA in no benchmark.
>
> You have raised an important point. One key feature of SNNs is using binary activation. Researches on quantization neural networks (QNNs) have shown that the binary activation function used by SNNs will degrade the performance to a large extent [1][2][3][4], which is even more than the influence of low-bit weights [3]. The relatively low performance of SNNs might originate from this.
>
> In addition, training deep SNNs is a new and developing research topic with fewer researchers compared with training analog neural networks (ANNs). As a result, it is not surprising that SNNs have relatively low performance compared with ANNs. Due to this reason, developing new learning schemes for SNNs is necessary, which is the aim of our work.
>
>
>
>     [1] Kim, H., Kim, K., Kim, J., & Kim, J. J. (2020). Binaryduo: Reducing gradient mismatch in binary activation network by coupling binary activations. arXiv preprint arXiv:2002.06517.
>     [2] Cai, Z., He, X., Sun, J., & Vasconcelos, N. (2017). Deep learning with low precision by half-wave gaussian quantization. In Proceedings of the IEEE conference on computer vision and pattern recognition (pp. 5918-5926).
>     [3] Mishra, A., Nurvitadhi, E., Cook, J. J., & Marr, D. (2017). WRPN: Wide reduced-precision networks. arXiv preprint arXiv:1709.01134.
>     [4] Zhou, S., Wu, Y., Ni, Z., Zhou, X., Wen, H., & Zou, Y. (2016). Dorefa-net: Training low bitwidth convolutional neural networks with low bitwidth gradients. arXiv preprint arXiv:1606.06160.
>
> > In line 233 you stated that more details about the configuration of training were available in the appendix, I was not able to find the appendix section, nor an appendix file on the supplement material, even though the code contains the configurations of the experiment.
>
> The appendix file is in the 'supplementary material.pdf'. The configuration of training is in 'Implementation Details' of the appendix.

---

### Official Review · Reviewer_1S4Q · 2022-07-12

**Rating:** 4
**Confidence:** 5
**Soundness:** 3 good
**Presentation:** 3 good
**Contribution:** 2 fair

**Summary:**

The authors propose a temporal Backprop approach for training SNNs with some interesting backward kernel fucntion.

**Questions:**

If the authors can highlight their technical novelty a compared to previous works, it will help me re-assess the paper's contributions. See above comments for reference.

**Limitations:**

See weakness section.

**Strengths And Weaknesses:**

+ The authors showcase that their temporal BP methodology yields high accuracy as compared to similar related works.
-The paper presents a direct training method using BP for SNNs. This work is very derivative and incremental. There is a lot of work from Priya Panda's group at Yale, Emre Neftci's group, and many others with regard to SNN training. The authors have failed to acknowledge most recent works and the method they are proposing is very incremental in the context of those works. Further, many recent works on SNNs have targeted larger datatsets including video segmenattion with direct training. I wonder if the author's method can even scale up, since their results are limited to CIFAR10, CIFAR100.

-The authors did not comment on how many time steps their method requires to train. In the recent work [5], the authors show that they can use backward connections to train SNNs better, is there a similarity between the backward kernel and backward conenction?


Below is a list of publications (not exhaustive) that the author should check:
[1]
[2] Enabling spike-based backpropagation for training deep neural network architectures C Lee, SS Sarwar, P Panda, G Srinivasan, K Roy Frontiers in neuroscience, 119
[3] Rate Coding Or Direct Coding: Which One Is Better For Accurate, Robust, And Energy-Efficient Spiking Neural Networks? Y Kim, H Park, A Moitra, A Bhattacharjee, Y Venkatesha, P Panda ICASSP 2022-2022
[4] Neuromorphic Data Augmentation for Training Spiking Neural Networks Y Li, Y Kim, H Park, T Geller, P Panda arXiv preprint arXiv:2203.06145
[5] Neural architecture search for spiking neural networks Y Kim, Y Li, H Park, Y Venkatesha, P Panda arXiv preprint arXiv:2201.10355
[6] Optimizing deeper spiking neural networks for dynamic vision sensing Y Kim, P Panda Neural Networks 144, 686-698
[7] Federated Learning with Spiking Neural Networks Y Venkatesha, Y Kim, L Tassiulas, P Pand IEEE Transactions on Signal Processing 2021
[8] Beyond classification: directly training spiking neural networks for semantic segmentation Y Kim, J Chough, P Panda arXiv preprint arXiv:2110.07742
[9] Visual explanations from spiking neural networks using interspike intervals Y Kim, P Panda Scientific Reports 11, Article number: 19037 (2021)
[10] Revisiting batch normalization for training low-latency deep spiking neural networks from scratch Y Kim, P Panda Frontiers in neuroscience, 1638

---

> ### Author Response · Authors · 2022-08-02
> **Response to Reviewer 1S4Q (Part 1)**
>
> We appreciate your constructive comments. We would like to address your concerns below.
>
> > If the authors can highlight their technical novelty and compare it to previous works, it will help me re-assess the paper's contributions.
>
> We would like to emphasize that our work focus on event-driven learning (with temporal gradient), which is different from the commonly used RNN-like BPTT learning approaches (with activation-based gradient). In our method, the gradient information is carried by spikes instead of both spikes and non-spikes at each time step (shown in Figure 1). This feature makes our method harder to train (because of the sparser gradient propagation path) as well as more biological plausibility.
>
> The main contributions of our paper are re-summarized as follows:
> 1. We prove that the typical SNN temporal backpropagation training approach assigns the gradient of an output spike of a neuron to the input spikes generating it. After summing this assignment rule altogether, we find that the sum of gradients is kept unchanged between layers.
> 2. We analyze the case of the pooling layer (which does not have weights) and find that average pooling does not keep the gradient sum unchanged, but we can modify its backward formulas to meet the requirement. Meanwhile, the max-pooling layer satisfies the rule initially.
> 3. We point out the reverse gradient problem in event-driven learning that the direction of the temporal gradient is reversed during backpropagation when the kernel function of an input spike is decreasing. Then we propose a backward kernel function that addresses this problem while keeping the sum of gradients unchanged between layers.
> 4. The adjusted average pooling layer and the non-decreasing backward kernel enhances the performance of our model as well as the convergence speed. To our best knowledge, our proposed approach achieves state-of-the-art performance on CIFAR10 among event-driven training methods (with temporal gradients) for SNNs. Meanwhile, our method is the first event-driven backpropagation approach that successfully trains SNN on the larger-scale CIFAR100 dataset.
>
> > The authors have failed to acknowledge most recent works. Below is a list of publications (not exhaustive) that the author should check.
>
> Thanks for recommending these works. We are encouraged to see that SNNs can be applied in such a variety of tasks. We have checked and added the following references [1-6] in the revised paper (please refer to the sections of Introduction and Backgrounds and Related Work). We will incorporate the discussion on these works in our final version, where the one additional page allows us to extend the current Sections with more content and illustration.
>
>
>     [1] Lee, C., Sarwar, S. S., Panda, P., Srinivasan, G., & Roy, K. (2020). Enabling spike-based backpropagation for training deep neural network architectures. Frontiers in neuroscience, 119.
>     [2] Kim, Y., & Panda, P. (2020). Revisiting batch normalization for training low-latency deep spiking neural networks from scratch. Frontiers in neuroscience, 1638.
>     [3] Kim, Y., & Panda, P. (2021). Optimizing deeper spiking neural networks for dynamic vision sensing. Neural Networks, 144, 686-698.
>     [4] Venkatesha, Y., Kim, Y., Tassiulas, L., & Panda, P. (2021). Federated learning with spiking neural networks. IEEE Transactions on Signal Processing, 69, 6183-6194.
>     [5] Kim, Y., Chough, J., & Panda, P. (2021). Beyond classification: directly training spiking neural networks for semantic segmentation. arXiv preprint arXiv:2110.07742.
>     [6] Kim, Y., Venkatesha, Y., & Panda, P. (2022). PrivateSNN: Privacy-Preserving Spiking Neural Networks. In Proceedings of the AAAI Conference on Artificial Intelligence (Vol. 36, No. 1, pp. 1192-1200).

---

> > ### Author Response · Authors · 2022-08-02
> > **Response to Reviewer 1S4Q (Part 2)**
> >
> > > I wonder if the author's method can even scale up since their results are limited to CIFAR10 and CIFAR100.
> >
> > We would like to note that the most complex dataset in which previous works successfully trained SNNs in **event-driven fashion** is the CIFAR10 dataset. Our work makes one step further and successfully trains SNNs on the CIFAR100 dataset. To the best of our knowledge, this is the first time that the time-based backpropagation approach successfully trains SNN on the CIFAR100 dataset. We are glad to investigate more complex datasets (such as TinyImageNet and ImageNet) further in future work.
> >
> >
> > > The authors did not comment on how many time steps their method requires to train.
> >
> > Thanks for your comments. We would like to note that training networks in an event-driven fashion do not necessarily need the concept of ‘time step’. As illustrated in Figure 1, we only have to record information at spike times (the precise timing, slope of membrane potential, etc.) to train our network, and there is no need to use clock-driven methods by nature. However, to better make use of the current deep learning frameworks, we turn the training process from continuous time to discrete time steps in our simulation. The number of time steps is set to 5 for MNIST and Fashion-MNIST, 12 for CIFAR10, 16 for CIFAR100, and 30 for N-MNIST. For further information about hyper-parameters, please refer to the appendix (section ‘Implementation Details’).
> >
> >
> > > In the recent work [5], the authors show that they can use backward connections to train SNNs better, is there a similarity between the backward kernel and backward connection?
> > > [5] Neural architecture search for spiking neural networks Y Kim, Y Li, H Park, Y Venkatesha, P Panda arXiv preprint arXiv:2201.10355
> >
> > We would like to clarify that the backward kernel we proposed differs from the backward connection. The backward connection in [5] is applied in both forward and backward propagation (it adds the transformed node feature of $l$-th layer at time-step $t−1$ to the node of $l′$-th ($l′<l$) layer at time-step t). In contrast, our backward kernel can be viewed as a correction of the original workflow in the backward propagation, which means it is only applied in the backward propagation.

---

> ### Author Response · Authors · 2022-08-06
> **Thank you for the time and hope our responses helpful for your re-assessment of our work.**
>
> Dear reviewer 1S4Q, we sincerely hope our posted response can help to address your concerns on our paper and serve as a reference for your re-assessment of our work. If you have any further comments and questions, please let us know and we are glad to write a follow-up response.

---

> ### Author Response · Authors · 2022-08-09
> **Restatement of our contribution that we hope can help for your re-assessment**
>
> Dear Reviewer 1S4Q,
>
> As you suggested, we have checked and added recommended publications in the new version of our paper. We organize the other concerns as follows:
> 1) Aiming at your question on the contribution of our paper (which is also asked by Reviewer tMmb), we have clarified the contribution of our paper in the response below.
> 2) For whether we can scale up our method, we have explained that we are the first to train SNNs in an event-driven fashion on the CIFAR100 dataset.
> 3) For your concern on the number of time steps, we have provided it in our response and other hyper-parameters in the supplementary material.
> 4) We have explained the difference between the backward connection in [1] and the backward kernel we used in our response.
>
> We kindly hope you to re-consider your initial rating based on the response we provided. If you have any further questions, please let us know and we are glad to provide a follow-up response.
>
> > [1] Neural architecture search for spiking neural networks Y Kim, Y Li, H Park, Y Venkatesha, P Panda arXiv preprint arXiv:2201.10355

---

### Official Review · Reviewer_kF1o · 2022-07-14

**Rating:** 7
**Confidence:** 4
**Soundness:** 3 good
**Presentation:** 4 excellent
**Contribution:** 3 good

**Summary:**

This paper focuses on the temporal, event-driven manners of training a spiking neural network from scratch. The authors first revisit the learning dynamics of event-driven learning and discover several invariance properties. Then, a problem called reverse gradient is raised and addressed. Extensive experiments are conducted to verify the effectiveness of this method.

**Questions:**

TBH, I am not an expert in event-based training of SNN, therefore I cannot give useful feedback with respect to that. I have a few questions about the difference between event-based training and the "RNN-like" training.

1. When implementing these event-based learning algorithms on the neuromorphic hardware, how to accelerate the training and why is it significantly faster than the RNN-based learning algorithms?

2. Can this method be trained with residual connection?


**Limitations:**

Overall, I found this paper interesting and could be valuable for publication at the NeurIPS conference. As I am familiar with this type of training method for SNNs, I could not give conceptual limitations for this paper. I am giving borderline acceptance of this paper due to its good presentation. Meanwhile, I will set my confidence score to 2 and will look into the comments from other reviewers to finalize my rating.


-----

POST-REBUTTAL REVIEW:

I'd like to thank the authors for their detailed response. My questions are addressed, thus I increase my rating to 7.


**Strengths And Weaknesses:**

1. This paper is well-presented. The writing and visualization are neat and easy to follow.

2. The experimental results are sufficient for comparison with existing event-driven learning works.

---

> ### Author Response · Authors · 2022-08-02
> **Response to Reviewer kF1o**
>
> Thank you for your positive and thoughtful comments. We are encouraged that you find our paper well written and could be valuable for publication at the NeurIPS conference. We would like to address your concerns and answer your questions in the following.
>
> > When implementing these event-based learning algorithms on neuromorphic hardware, how to accelerate the training and why is it significantly faster than the RNN-based learning algorithms?
>
> You have addressed an interesting concern about running our algorithm on neuromorphic hardware. We have discussed this with some hardware experts, and they suggest that accelerating the network on hardware can be solved by representing spike inputs and outputs in a compressed, time-stamped, and sorted way, then sequentially walking through time-stamped spikes and avoiding calculating when spikes are not emitted in simulation [1].
>
> Next, we show that the number of operations (total times all neurons affected by spikes) of the event-based learning algorithms is less than the RNN-based learning algorithms when spikes are sparse. For simplicity, we only analyze a single fully-connected layer with M input neurons and N output neurons. Other layers and the whole network can be analyzed similarly.
> During training and inference, RNN-based learning algorithms are forced to unfold through the time axis, as explained in Figure 1 and Section 2. As a result, the corresponding number of operations is at least $O(TMN)$, where T is the total time steps and M, N are the number of input and output neurons. On the other hand, event-based learning algorithms only have to deal with cases where a certain neuron fires a spike, and record the relevant information. In the forward stage, a spike fired by an input neuron affects the state itself and all output neurons, which is $O(N)$ in total. In the backward stage, a spike fired by an output neuron needs to propagate gradient information to all spikes between this spike and the last spike fired by this neuron (so all input spikes are processed once in this stage). Therefore, denoting the average firing rate of input and output neurons to be $\alpha$ and $\beta$, the number of operations of this layer is $O(T(\alpha MN+\beta M+\alpha N))=O(T(\alpha MN+\beta M))$. When spikes are sparse, event-based learning algorithms certainly have advantages since $\alpha+\beta \ll 1$ in this case.
> We have added the analysis in the appendix of the revised paper.
>
>
>
>     [1] Narayanan, S., Taht, K., Balasubramonian, R., Giacomin, E., & Gaillardon, P. E. (2020, May). SpinalFlow: An architecture and dataflow tailored for spiking neural networks. In 2020 ACM/IEEE 47th Annual International Symposium on Computer Architecture (ISCA) (pp. 349-362). IEEE.
>
>
> > Can this method be trained with residual connection?
>
> You have raised an interesting concern. We test ResNet on CIFAR10 and find that it performs even better than the results presented in our paper (92.45% by a 14-layer Spike-Element-Wise (SEW) ResNet).
> One thing that is worth noticing is the case when two simultaneous input spikes are added in the residual connection. According to our analysis in 'Invariant sum of gradients among layers with weights' in section 3.2, the gradients of input spikes 'inherit' gradients from the next output spike and keep the sum of gradients unchanged. However, in the add operation, the default gradient rule in adding is to copy the gradient from the added spike to the two input spikes (since when $s=a+b$, $\frac{\partial s}{\partial a} = \frac{\partial s}{\partial b} = 1$). This causes gradients amplified in previous layers, which can be corrected by a custom backward function.
> We have added the results in Table 2 of the revised paper.

---

### Meta-Review · Area_Chair_e9Nv · 2022-08-24

**Recommendation:** Accept
**Confidence:** Less certain

**Metareview:**

The authors propose a novel training algorithm to train spiking neural networks (SNNs) in an event-driven manner with backpropagation. They perform experiments on standard benchmarks such as CIFAR-10 and CIFAR-100 to verify the effectiveness of the method. The algorithm achieves SOTA performance on these data sets. Event-driven methods are interesting from a hardware perspective as gradient have to propagated only at spike times.

The manuscript received mixed ratings, a clear agreement could not be found.

Pros:
- The authors provide an analysis of event-driven backprop in SNNs which helps to adjust the usual learning procedure.
- The authors performed experiments on several data sets and achieved SOA performance w.r.t. other event-driven methods.
- They also tackled CIFAR100 for which no results were previously shown with event-driven algorithms
- The paper is well-written, although language could be improved at places.

Cons:
- Improvements over competing event-driven techniques are rather small
- Performance is still clearly below non-event based surrogate gradient methods (but this is not surprising)
- Not clear how the method scales up beyond CIFAR100

Since the ratings were mixed, I read the paper and believe it is publishable in NeurIPS although it is somewhat borderline.

**Award:**

No

---

### Decision · Program_Chairs · 2022-09-14

Accept